# CPAP Influence on Readily Available Inflammatory Markers in OSA—A Pilot Study

**DOI:** 10.3390/ijms232012431

**Published:** 2022-10-17

**Authors:** Ioana Madalina Zota, Cristina Andreea Adam, Dragoș Traian Marius Marcu, Cristian Stătescu, Radu Sascău, Larisa Anghel, Daniela Boișteanu, Mihai Roca, Corina Lucia Dima Cozma, Alexandra Maștaleru, Maria Magdalena Leon Constantin, Elena Andreea Moaleș, Florin Mitu

**Affiliations:** 1Department of Medical Specialties (I), Faculty of Medicine, University of Medicine and Pharmacy “Grigore T. Popa”, 700115 Iași, Romania; 2Department of Medical Specialties (III), Faculty of Medicine, University of Medicine and Pharmacy “Grigore T. Popa”, 700115 Iași, Romania

**Keywords:** obstructive sleep apnea (OSA), continuous positive airway pressure (CPAP), inflammation, C-reactive protein (CRP), red cell distribution width (RDW), neutrophil-to-lymphocyte ratio (NLR), platelet-to-lymphocyte ratio (PLR)

## Abstract

Obstructive sleep apnea (OSA) is characterized by repetitive upper airway collapse, chronic hypoxia and a proinflammatory phenotype. The purpose of our study was to evaluate readily available inflammatory biomarkers (C-reactive protein (CRP), erythrocyte sedimentation rate (ESR), white blood cell count (WBC), red cell distribution width (RDW), neutrophil-to-lymphocyte ratio (NLR), platelet-to-lymphocyte ratio (PLR), mean platelet volume (MPV), WBC-to-MPV ratio (WMR) and lymphocyte-to-C-reactive protein ratio (LCR)) before and after CPAP in patients with moderate–severe OSA. We performed a prospective study that included patients with newly-diagnosed moderate–severe OSA. The control groups (patients without OSA and with mild OSA) were selected from the hospital polygraphy database. All subjects underwent routine blood panel, which was repeated in moderate–severe OSA patients after 8 weeks of CPAP. Our final study group included 31 controls, 33 patients with mild, 22 patients with moderate and 37 patients with severe OSA. CRP, ESR, NLR and WMR were correlated with OSA severity. After 8-week CPAP therapy, we documented a decrease in weight status, which remained statistically significant in both CPAP-adherent and non-adherent subgroups. Readily available, inexpensive inflammatory parameters can predict the presence of moderate–severe OSA, but are not influenced by short-term CPAP.

## 1. Introduction

Obstructive sleep apnea (OSA) is the most common sleep-related disorder, with a recently reported marked rise in prevalence [1,2]. The hallmark of obstructive sleep apnea (OSA) is recurrent complete or partial upper airway collapse during sleep, which causes repetitive microawakenings, chronic hypoxia, hypercapnia and oxidative stress, thus promoting a systemic proinflammatory phenotype [3]. The chronic, subclinical pro-inflammatory status is maintained by the frequent association of OSA with obesity, type 2 diabetes mellitus and non-alcoholic fatty liver disease [4,5]. Abdominal obesity is a robust predictor of OSA [6,7]. The adipose tissue is a true endocrine organ that responds to hypoxia by releasing resistine (that activates the NF-kB pathway and promotes the expression of proinflammatory cytokines and insulin resistance) and inflammatory cytokines (TNF-α, IL-6, IL-1B, IL-18) [4,7]. Several studies have reported that OSA patients present increased serum levels of inflammatory biomarkers, some of which correlate with OSA severity [6,8,9,10]. In addition, OSA patients present evidence of local, mucosal inflammation (mucosal inflammatory infiltrate), subepithelial oedema and elevated exhaled nitric oxide (NO) levels [11,12,13,14], partially explained by the irritative effect of repetitive airway collapse.

CRP, IL-6 and TNF-α are the most studied inflammatory markers in OSA and are elevated in OSA patients compared to controls [7]. RDW reflects red blood cells’ variability in size and volume and has been previously correlated with CRP and ESR [15]. RDW has been nominated as an index for estimating autoimmune disease activity [16], a marker of subclinical inflammation and underlying atherosclerosis [15] and of increased cardiovascular morbi-mortality [17]. Furthermore, as RDW correlates with several OSA severity parameters, it could be viewed as a rudimentary OSA screening tool in the general population [18].

Additionally, it seems that circulating neutrophils and thrombocytes present a proinflammatory and prothrombotic phenotype in OSA patients, which is partially reversible after CPAP [6]. Although platelets were primarily viewed as instruments of thrombosis and neutrophils mainly as inflammatory effectors, the in-depth study of neutrophil extracellular traps unveiled the fine interplay between platelets, neutrophils and lymphocytes in both inflammation and thrombosis [19]. While thrombocytes and granulocytes act as acute phase reactants and increase during infection and inflammation, a low lymphocyte count is an uncontrolled inflammatory pathway. The platelet-to-lymphocyte ratio (PLR), the neutrophil-to-lymphocyte ratio (NLR) and the WMC-to-MPV ratio (WMR) integrate opposite inflammatory pathways and can easily be calculated from a standard complete blood count. These readily available inflammatory biomarkers have potential prognostic implications in malignancies [20,21], autoimmune [22], respiratory [23] and cardiovascular disease [24,25,26]. PLR seems to be more influenced by hypoxemia and has recently proved to be a poor, but significant predictor for OSA-COPD [27]. Larger platelets seem to have a prothrombotic phenotype and are associated with increased cardiovascular risk [28]. Mean platelet volume (MPV) is independently correlated with OSA severity [29,30] and decreases after medium-term CPAP [31]. While CPAP partially reverses cardiovascular and metabolic disturbances in OSA [32,33,34,35,36,37,38], its effect on local and systemic OSA-related inflammation has yielded conflicting results [17,39,40,41,42,43,44,45,46,47]. Overall, persistent systemic inflammation seems to be an important feature of OSA. We hypothesized that readily available, inexpensive inflammatory biomarkers could be a useful tool in assessing CPAP effectiveness and in predicting OSA severity.

## 2. Results

We included in our study a total of 123 subjects, 92 of whom were previously diagnosed with OSA, and 31 controls. Of the 92 patients diagnosed with OSA, 33 patients had mild, 22 patients moderate and 37 patients had severe OSA. Table 1 shows the demographic, anthropometric and biological parameters assessed in patients enrolled in the study.

In terms of demographic parameters, predominantly male patients were enrolled, with the most significant percentage of patients having a moderate form of OSA (60.6% vs. 72.6% vs. 70.3%). The average age of patients was higher in OSA patients compared to the control group, with the oldest patients being identified in the severe subgroup of patients (54.06 ± 15.37 vs. 57.68 ± 9.18 vs. 58.49 ± 9.49 years old). Age was a statistically significant parameter associated with moderate (*p* = 0.021) and severe (*p* = 0.003) forms of OSA.

Among the anthropometric parameters, special attention was given to the BMI, noting that its average value increased with OSA severity (32.34 ± 5.44 vs. 32.65 ± 6.16 vs. 35.41 ± 5.63 kgm^2^). Statistical analysis revealed a statistically significant correlation between BMI and severe form of OSA (*p* = 0.035).

In addition to the demographic and anthropometric data presented above, the statistical analysis also included hematological and biochemical parameters. In the case of WBC (6920.30 ± 1731.96/mm^3^ vs. 6322.73 ± 1719.62/mm^3^ vs. 6883.24 ± 1858.72/mm^3^) and RDW (12.86 ± 2.73 vs. 13.53 ± 1.03 vs. 13.94 ± 1.35) descriptive statistical analysis revealed mean serum values directly proportional to the clinical form of OSA. MPV (10.37 ± 1.06 vs. 9.03 ± 0.80 vs. 9.62 ± 0.96) and CRP (0.51 ± 0.42 mg/dl vs. 0.61 ± 0.43 mg/dl vs. 0.98 ± 1.30 mg/dl) recorded higher mean serum values among patients with mild OSA. NLR (1.90 ± 0.54 vs. 2.10 ± 0.75 vs. 2.34 ± 1.13), LCR (1632.83 ± 1421.37 vs. 11,318.78 ± 32,732.56 vs. 17,090.79 ± 53,940.05) and PLR (123.97 ± 36.05 vs. 127.79 ± 41.69 vs. 134.45 ± 58.31) had increasing mean serum values in a directly proportional way to the severity of OSA, but without statistical significance compared to controls.

In the entire study group, AHI was significantly correlated with NLR, WMR, RDW, MPV, ESR and CRP (Table 2, Figure 1 and Figure 2). However, subgroup analysis showed most statistically significant correlations remained significant only in patients with severe OSA—WMR (*p* < 0.001) and CRP (*p* = 0.019). PLR exhibited a positive strong association with OSA severity in controls (*p* < 0.001) and patients with mild OSA (*p* < 0.001), but a mild negative association in patients with severe OSA.

The average CPAP use in our moderate–severe OSA patients (59 patients) was 4.01 ± 2.21 h/night. Average CPAP use was 2.15 ± 1.14 h/night and 5.86 ± 1.24 h/night in the non-adherent and adherent subgroups, respectively. After 8-week CPAP therapy, we observed a statistically significant decrease in weight and BMI (Table 3), which remained statistically significant in both adherent and non-adherent subgroups. For patients with moderate and severe OSA, we used age, BMI and inflammatory parameters, and developed a statistically significant logistic regression model dependent on CPAP adherence (*p* = 0.010) However, none of the analyzed inflammatory biomarkers was statistically influenced by CPAP (Table 4 and Table 5).

A ROC curve analysis (Figure 3 and Figure 4) was performed to identify inflammatory parameters that can predict moderate–severe or severe OSA. LCR (area under curve <AUC> = 0.769, *p* < 0.001, cut-off value of 2886.79), WMR (AUC = 0.634, *p* = 0.021, cut-off value of 713.61), NLR (AUC = 0.639, *p* = 0.016, cut-off value of 2.10) and RDW (AUC = 0.667, *p* = 0.004, cut-off value of 13.65) were associated with the presence of moderate–severe OSA (AHI ≥ 15). The predictive value of these parameters decreases for severe OSA (AHI ≥ 30), the most significant in this case being NLR (AUC = 0.631, *p* = 0.031, cut-off value of 2.22), RDW (AUC = 0.637, *p* = 0.024, cut-off value of 13.55) and LCR (AUC = 0.662, *p* < 0.008, cut-off value of 2429.02).

An additional ROC curve analysis (Figure 5 and Figure 6) was performed to identify inflammatory parameters that could predict CPAP adherence in OSA patients. Thus, in patients with moderate OSA, statistical analysis revealed LCR (area under curve <AUC> = 0.825, *p* = 0.010), PLR (AUC = 0.800, *p* = 0.018) and CRP (AUC = 0.083, *p* = 0.001) as inflammatory biomarkers associated with CPAP adherence. In severe OSA, the predictive value of these analyzed inflammatory parameters did not reach statistical significance.

## 3. Discussion

Inflammation plays a pivotal role in the physiopathology of OSA, promoting endothelial dysfunction, accelerated atherosclerosis and thrombotic complications, thus increasing cardiovascular morbimortality [4,6]. The adipose tissue is a true endocrine organ that responds to hypoxia by releasing adipocytokines [4]. Circulating monocytes, neutrophils and thrombocytes present proinflammatory and prothrombotic features in OSA patients, partially reversed after CPAP [6]. Cytotoxic T cells also exhibit a proinflammatory phenotype and release higher quantities of tumor necrosis factor α (TNF α) and interleukin 8 (IL-8), to the detriment of anti-inflammatory mediators (IL-10) [6]. Indeed, several studies have reported that OSA patients present increased serum levels of proinflammatory cytokines and acute phase proteins, some of which correlate with OSA severity [6,8,9,10]. Routine inflammatory biomarkers are useful in estimating long-term cardiovascular risk, but reported baseline values present significant variations between studies, due to differences in selection criteria, blood sampling timing and biochemical analysis protocols [48,49,50].

The elevated inflammatory biomarkers observed in OSA could be explained by the presence of associated cardiometabolic comorbidities [4,5]. However, Vicente et al. [39] previously showed that pharyngeal lavage (but not serum) inflammatory markers are increased in comorbidity-free OSA cases, and that they decrease after 1 year of treatment (CPAP or surgery) [39].

Most of the literature results regarding the benefit of CPAP in reducing inflammation have yielded controversial results [4,14]. Although three meta-analyses of case-control studies have documented a significant reduction in IL-8, CRP and TNF-α [45,46,47], most randomized control trials (RCTs) failed to demonstrate a significant impact of CPAP initiation or withdrawal upon inflammatory biomarkers [4,40,41,42]. However, one RCT reported a decrease in tumor necrosis factor receptor 1 (TNFR-1) after 12 weeks of CPAP [51]. Although this result was not confirmed by a subsequent trial (Kritikou et al. [52], 2 months of CPAP), the hypothesis remains intriguing, as TNFR-1 is a highly sensitive inflammatory marker that plays a significant role in glucose homeostasis and the development of obesity [4].

### 3.1. ESR and CRP

ESR and CRP, the two ”basic” inflammatory markers, are subject to numerous confounding factors, which could explain why their study in different OSA populations has provided divergent findings. In our analysis, contrary to the results of Lee et al. [53], ESR did not significantly vary with OSA severity. Although ESR was correlated with AHI, it did not predict the presence of OSA or CPAP adherence. Furthermore, ESR was not influenced by short-term CPAP therapy.

In line with previous reports [17,43,54], our results show that CRP is higher in patients with OSA. However, in the study by Kurt et al. [55], CRP did not significantly vary with OSA severity. CRP was positively correlated with AHI only in severe OSA patients, and surprisingly presented a negative correlation with AHI in the study group as a whole (*n* = 123). CRP was not associated with the presence of moderate–severe OSA but predicted CPAP adherence in moderate OSA patients.

Although most of the literature reports suggest a clear association between CRP and OSA, our analysis, as well as previous randomized control trials [40,41,42], show that CRP does not decrease after short- and moderate-term CPAP. The physiopathology of elevated CRP levels in OSA patients is multifactorial physiopathology and should be addressed accordingly—intensive statin therapy, optimal glycemic control and aggressive correction of cardiovascular risk factors (including weight management and smoking cessation). In fact, Chirinos et al. [40] showed that weight-loss interventions were associated with a significant improvement in CRP [40]. It should be noted that while a statistically significant weight loss was observed in all our study groups, its amplitude was mild and it had no clinical impact. Overall, it seems that CRP and ESR have little value in predicting the presence of OSA.

### 3.2. RDW

RDW is a cheap and readily available parameter that reflects red blood cells variability in size and volume and is not influenced by sex, age and BMI [17]. RDW has been previously correlated with CRP and ESR values [15] and has been nominated as an index for estimating autoimmune disease activity [16] and a marker of subclinical inflammation, underlying atherosclerosis [15] and cardiovascular morbi-mortality [17]. OSA-induced repetitive hypoxia promotes accelerated erythropoiesis [43] which could explain a higher RDW in OSA [15,17,18,43,56]. Durmaz et al. [57] have previously shown that RDW is a predictor of OSA burden in severe OSA patients. RDW values were previously correlated with AHI, DI, minimal nocturnal O2Sa and ESS results [18,58,59,60,61], suggesting that RDW could be used as an OSA screening tool in the general population. However, CPAP therapy seems to have little impact on RDW values [17,43,44]: while Leon Subias et al. reported no changes in RDW after 1 year of CPAP [17], two other studies surprisingly reported an increase in RDW after 3–6 months of CPAP [43,62].

In our analysis, RDW was significantly higher in patients with severe OSA compared to controls, and an NLR > 13.65 predicted the presence of moderate–severe OSA (AUC = 0.667, *p* = 0.004). In the entire group of patients (*n* = 123), RDW was mildly but significantly associated with AHI (r = 0.2, *p* = 0.01), but the association lost its statistical significance in subgroup analysis. Furthermore, RDW did not predict adherence and we recorded no significant change in RDW after 8 weeks of CPAP. As such, RDW could be used to prioritize polysomnographic evaluations in patients with a clinical suspicion of OSA and should be considered as a marker of high cardiovascular risk in sleep apnea [59,60].

### 3.3. MPV

Platelet activation, adhesion and aggregation occur in all inflammatory states, and are associated with an increased platelet volume [63,64]. Previous studies have documented increased platelet activation and aggregation in OSA patients [65,66,67], which is partially reversed by CPAP [66]. Since larger platelets seem to have a prothrombotic phenotype and are associated with the development of cardiovascular complications [28], MPV was proposed as a novel predictor of atherosclerosis [68,69]. Two previous studies showed that MPV is elevated in severe OSA [70,71] and a recent meta-analysis reported gradually increasing MPV levels with increasing OSA severity [61]. However, only half of the 8 available studies found a significant difference regarding MPV in OSA versus controls [29,30,43,60,63,64,72,73], and Topçuoğlu et al. [64] showed that MPV is not a severity indicator in comorbidity-free OSA patients. While three studies found an independent correlation between MPV and OSA parameters (ESS score, AHI, average SpO2, min SpO2 and % of total sleep time with SpO2 < 90%) [29,30,71], MPV was not significantly correlated with AHI in another report [55]. CPAP seems to have an inconsistent effect on mean platelet volume—MPV decreased after 6 months of CPAP in severe OSA [31,43], remained unchanged in another report [74] and increased after 3 months in a small group of 29 comorbidity-free OSA patients [62].

Our analysis surprisingly showed lower MPV values in patients with moderate and severe OSA compared to controls. Furthermore, MPV was negatively correlated with AHI (r = −0.15, *p* = 0.01) and did not predict CPAP adherence or the presence of moderate–severe OSA. Although MPV mildly decreased after 8 weeks of CPAP, the difference did not reach statistical significance (*p* = 0.16). Our results suggest that MPV has little value in the assessment of OSA patients.

### 3.4. PLR

PLR integrates opposite inflammatory pathways and has potential prognostic implications in autoimmune [22], respiratory [23] and cardiovascular disease [24,25,26]. Hypoxemia, more than inflammation, seems to have a stronger impact on PLR, which has a better discriminative value in patients with OSA or chronic obstructive pulmonary disease (COPD). A recent report showed that PLR is a poor, but significant predictor for OSA-COPD [27]. Another study documented a significant association between AHI and PLR and revealed that a PLR value beyond 159 is independently associated with the presence of hypertension in OSA patients [75]. Although a recent meta-analysis reported that PLR gradually increases with OSA severity [61], two case-control studies [60,76] found no significant differences in PLR between controls and OSA, and another study reported lower PLR values in controls versus patients with sleep apnea [77].

In our analysis, PLR did not significantly vary between controls and patients with mild, moderate and severe OSA. PLR exhibited a strong, positive correlation with AHI in controls and patients with mild OSA, but exhibited a mild negative correlation with AHI in patients with severe sleep apnea. Our ROC curves showed that PLR was not associated with the presence of OSA, and predicted CPAP adherence only in moderate OSA patients. Furthermore, in line with the results of Ozdemir et al. [62], PLR was not significantly influenced after short-term CPAP. Altogether, PLR seems to have little value in the assessment and follow-up of OSA patients.

### 3.5. NLR

Intermittent hypoxia upregulates NF-κB, which in turn activates both granulocyte and macrophage colony-stimulating factor genes, explaining the association between NLR and OSA [78]. A previous report suggested that NLR could be used as a marker of chronic hypoxic burden in OSA, and therefore as a readily available parameter to evaluate CPAP effectiveness [77]. Furthermore, NLR was associated with cardiovascular comorbidities in OSA patients [79,80] and was an independent predictor of OSA in three other studies [54,81,82]. In a multi-center retrospective study (481 OSA cases and 80 controls) NLR was higher in patients with severe OSA, compared to controls and moderate and mild OSA [83], and a recent meta-analysis [84] concluded that neutrophilic inflammation plays a key role in OSA pathogenesis, suggesting that NLR could be viewed as a disease activity biomarker.

Four previous reports found mild, but statistically significant correlations between NLR and AHI [54,79,81,85], and several prospective studies documented a decrease in NLR after OSA treatment (1 month with a mandibular advancement device [86], 3 months of CPAP [81,87]). On the contrary, Ozdemir et al. [62], failed to document a significant impact of CPAP on NLR.

In our study, although NLR was mildly associated with AHI in the study group as a whole, the correlation did not reach statistical significance in subgroup analysis. Furthermore, NLR did not decrease after 8 weeks of CPAP. However, an NLR value > 2.1 predicted the presence of moderate–severe OSA (AUC = 0.639, *p* = 0.016), which should be taken into consideration in the initial evaluation of patients with a clinical suspicion of OSA. As obesity is a well-recognized confounder of NLR [84,88], the value of this parameter in lean OSA patients should be addressed in future clinical research.

### 3.6. WMR

WMR is another inflammatory marker with potential predictive value for the presence of severe OSA [76]. WMR is easy to calculate, reproducible, simple to use and with superior stability and lower variability to external cofounders compared to other inflammatory parameters. In our study, WMR showed a strong, significant correlation with AHI in patients with severe OSA (r = 0.603, *p* < 0.001). Furthermore, WMR was a significant predictor of the presence of moderate–severe OSA (AUC = 0.634, *p* = 0.021, cut-off value of 713.61). This is the first study to report the impact of CPAP on WMR. Although 8 weeks of CPAP did not significantly influence WMR, due to the aforementioned association between WMR and severe OSA, further studies should address the effect of medium and long-term CPAP on this readily available inflammatory biomarker.

### 3.7. LCR

LCR is an inflammatory biomarker used for indirect assessment of systemic inflammation. A recent study attributed to LCR a diagnostic and prognostic role in OSA patients [89]. Although our analysis did not document a significant correlation between AHI and LCR, a cut-off value of 2886.79 for LCR was associated with the presence of moderate–severe OSA (area under curve <AUC> = 0.769, *p* < 0.001). LCR also predicted CPAP adherence in patients with moderate sleep apnea. This is the first study to assess the impact of CPAP on LCR. The dynamic assessment performed 8 weeks after the start of CPAP therapy showed a reduction in mean serum levels that did not reach statistical significance. In summary, LCR could be used to prioritize polysomnographic evaluations in patients with a clinical suspicion of OSA. Its value in predicting CPAP adherence requires further studies. Despite our current findings, given the strong association between LCR and severe OSA, we consider that long-term CPAP could positively impact LCR, a hypothesis that deserves to be addressed by further research.

This study has several limitations. Most importantly, this was not a randomized trial, it had a short follow-up duration and our patients presented only borderline CPAP adherence. The controls and OSA subgroups were inhomogeneous in terms of the number of participants, age and gender, and were not BMI-matched. However, this is a reflection of daily practice, as patients with clinically significant OSA tend to be more obese and older. Our analysis included a larger representation of patients with moderate–severe OSA, which reflects the usual distribution of patients addressed for sleep studies. Active smoking can influence the WBC count, but it had a relatively low prevalence in our OSA patients. The presence of young platelets and protein disulfide isomerase are other potential confounders that were not taken into consideration in our study.

Although persistent systemic inflammation is an important feature of OSA, the impact of CPAP on inflammatory biomarkers remains controversial [44]. Our study, as well as other reports [17,43,62,90], did not feature significant changes in inflammation markers after 2 months CPAP. Inflammatory status is subject to the influence of genetic and environmental factors, and individual lifestyle, which partly explains the divergent results regarding the effectiveness of CPAP in reducing inflammatory markers. As chronic inflammation promotes endothelial dysfunction, accelerated atherosclerosis and thrombotic complications [6], the analysis of readily available, inexpensive inflammatory parameters could be extremely valuable in the screening of patients at high risk of developing target organ complications [9]. Basic blood count indices (LCR, WMR, NLR and RDW) could distinguish patients at increased cardiovascular risk and help prioritize polysomnographic evaluations in patients with a clinical suspicion of OSA.

## 4. Materials and Methods

### 4.1. Study Design

Patients with newly diagnosed moderate or severe OSA (prior to the initiation of CPAP therapy) were prospectively recruited in the 3rd Pneumology Clinic in Iași (January to December 2018). We excluded patients with previous CPAP therapy, central sleep apnea, non-OSA primary sleep disorder, anemia, chronic inflammatory disease, malignancy, chronic alcoholism, acute medical conditions in the prior 30 days, class IV heart failure, stage 5 chronic kidney disease, and Child–Pugh B and C cirrhosis. Eligible patients were clinically and biologically evaluated in the Cardiovascular Rehabilitation Clinic in Iași, before and after 8 weeks of CPAP. The control groups (patients with mild OSA and patients without OSA) were selected from the Clinical Rehabilitation Hospital cardiorespiratory polygraphy database (Philips Respironics Alice Night One).

### 4.2. OSA Diagnosis and Treatment

OSA was diagnosed by ambulatory or in-hospital six-channel cardiorespiratory polygraphy (Philips Respironics Alice Night One or DeVilbiss Porti 7). The recordings were manually scored by experienced sleep physiologists, according to the third International Classification of Sleep Disorders criteria [91]. Apnea was defined as a ≥90% reduction in oro-nasal airflow for at least 10 s. Hypopnea was defined as a ≥30% reduction in oro-nasal airflow for at least 10 s, associated with a ≥3% oxygen desaturation. CPAP effective pressure autotitration in the sleep laboratory was determined using a REMstar Auto C-Flex CPAP (Philips Respironics, Murrysville, PA, USA), a DreamStation Auto CPAP (Philips Respironics, Murrysville, PA, USA) or an AirSense 10 Autoset CPAP (ResMed, San Diego, CA, USA). Moderate and severe OSA was diagnosed as having an apnea–hypopnea index (AHI) of 15–30 events/h and >30 events/h, respectively. Follow-up cardiorespiratory polygraphy data was not performed due to the short follow-up of patients (8 weeks).

OSA patients received standard CPAP therapy with a REMstar Auto C-Flex CPAP (Philips Respironics, Murrysville, PA, USA), a DreamStation Auto CPAP (Philips Respironics, Murrysville, PA, USA) or an AirSense 10 AutoSet CPAP (ResMed, San Diego, CA, USA). OSA patients were reevaluated at the same clinic, 8 weeks after initiating CPAP therapy. After assessing CPAP adherence (at the 8-week follow-up), we divided our initial study population into two subgroups: adherent and nonadherent patients. Adherence was defined as a device usage time ≥ 4 h/night, while nonadherence was defined as a CPAP usage time < 4 h/night [92]. 

### 4.3. Measurements

Following hospital protocol, blood samples were collected a jeun, in the morning upon admission to the Cardiovascular Rehabilitation Clinic. All blood samples were processed in the hospital’s laboratory using a Pentra DF Nexus Hematology System^®^ (Horiba Healthcare, Kyoto, Japan) for complete blood count and a Transasia XL 1000 Fully Automated Biochemistry Analyzer (Transasia Bio-Medicals Ltd., Mumbai, India) for biochemistry. We recorded the following inflammatory biomarkers: erythrocyte sedimentation rate, red cell distribution width, mean platelet volume and C-reactive protein (CRP). NLR was calculated using the absolute neutrophil (N) and lymphocyte (L) values, using the following formula: NLR = N/L. PLR was calculated using the absolute platelets (P) and lymphocyte (L) values, using the following formula: PLR = P/L. WMR was calculated using the absolute WBC and MPV values, using the following formula: WMR = WBC/MPV. LCR ratio was calculated using the absolute lymphocyte (L) and C-reactive protein values, using the following formula: LCR = L/CRP.

CPAP adherence data (device usage, hours per night at the prescribed pressure) was recorded by the machine and downloaded using the appropriate software: EncoreBasic v.2.1 (Philips Respironics, Murrysville, PA, USA), Encore Pro 2 v.2.17 (Philips Respironics, Murrysville, PA, USA) or ResScan v.6.0 (ResMed, San Diego, CA, USA).

All anthropometric body measurements were performed three times. Height and weight were assessed in the morning upon admission, without shoes and with light clothing. Body mass index (BMI) was calculated as weight (kg)/height (m^2^). Waist circumference (WC) was measured horizontally at the top of the right iliac crest, at the end of a normal expiration.

### 4.4. Statistical Analysis

Data analysis was performed using SPSS 26.0 (Statistical Package for the Social Sciences, Chicago, IL, USA). For continuous data, the normality of distribution was assessed by Shapiro–Wilk test. Data are presented as mean ± standard deviation (SD) for continuous variables with normal distribution. An independent samples *t*-test was used to compare continuous variables with normal distribution. A non-parametric Mann–Whitney’s U test was applied to compare the variables not satisfying the assumption of normality. A multivariate logistic regression model was used to assess the independent predictors of OSA. Receiver operating characteristic (ROC) curve analysis was performed to determine the area under the curve for inflammatory parameters. Correlation between normally distributed parameters was assessed calculating Pearson correlation coefficients. A two-sided *p* value < 0.05 was considered significant for all analyses.

### 4.5. Ethics Statement

All patients signed a written informed consent form for inclusion. The study was conducted in accordance with the Declaration of Helsinki [93] and the protocol was approved by the Ethics Committee of the University of Medicine and Pharmacy “Grigore T. Popa” Iași (ethical approval code 1183/17.01.2018).

## 5. Conclusions

Some readily available, inexpensive inflammatory parameters can predict the presence of moderate–severe OSA and could prove helpful in OSA risk stratification, but are not influenced by short-term CPAP.

## Figures and Tables

**Figure 1 ijms-23-12431-f001:**
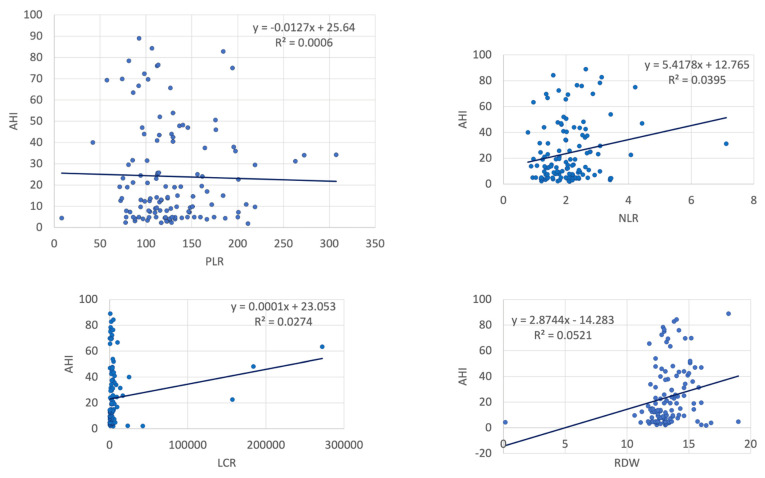
Correlations between AHI and inflammatory markers (123 patients, controls and OSA) (AHI: apnea–hypopnea index, NLR: neutrophil–lymphocyte ratio, LCR: lymphocyte-to-C-reactive protein ratio, RDW: red cell distribution width).

**Figure 2 ijms-23-12431-f002:**
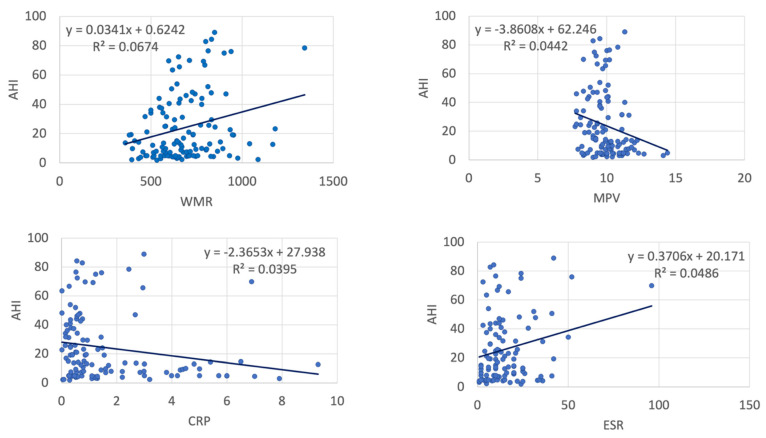
Correlations between AHI and inflammatory markers (123 patients, controls and OSA) (AHI: apnea–hypopnea index, MPV: mean platelet volume, CRP: C-reactive protein, ESR: erythrocyte sedimentation rate).

**Figure 3 ijms-23-12431-f003:**
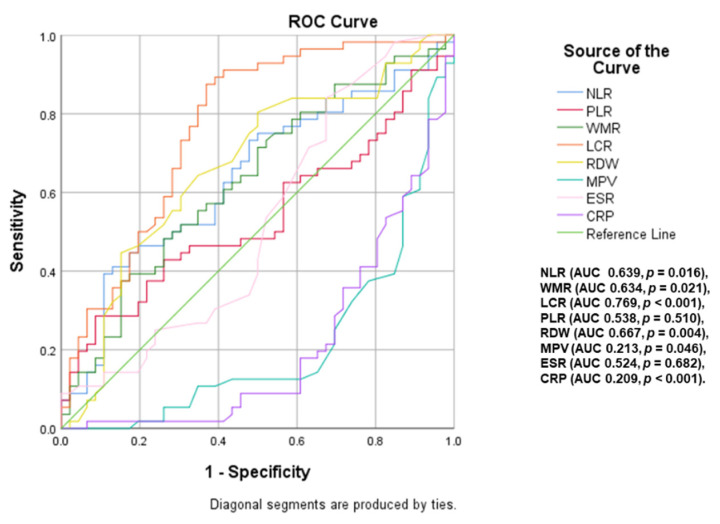
Impact of inflammatory parameters in predicting moderate–severe OSA (NLR: neutrophil-lymphocyte ratio; WMR: white blood cell count-to-mean platelet volume ratio; LCR: lymphocyte-to-CRP ratio; ESR: erythrocyte sedimentation rate; CRP: C-reactive protein; PLR: platelet-to-lymphocyte ratio; RDW: red cell distribution width; MPV: mean platelet volume).

**Figure 4 ijms-23-12431-f004:**
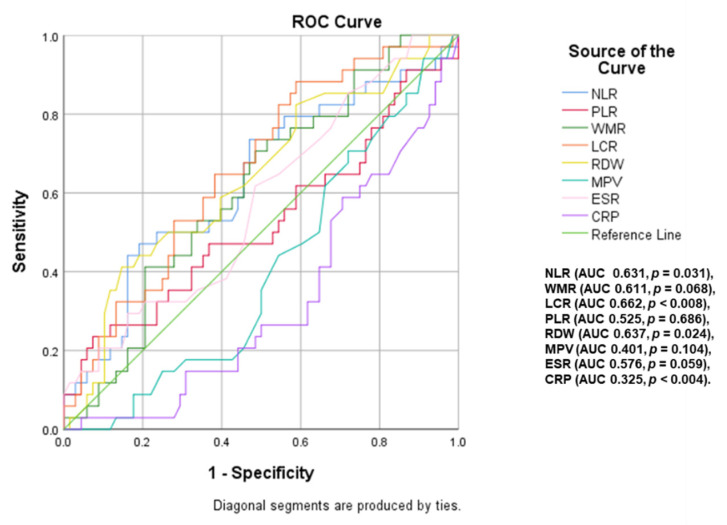
Impact of inflammatory parameters in predicting severe OSA (NLR: neutrophil–lymphocyte ratio; WMR: white blood cell count-to-mean platelet volume ratio; LCR: lymphocyte-to-CRP ratio; ESR: erythrocyte sedimentation rate; CRP: C-reactive protein; PLR: platelet-to-lymphocyte ratio; RDW: red cell distribution width; MPV: mean platelet volume).

**Figure 5 ijms-23-12431-f005:**
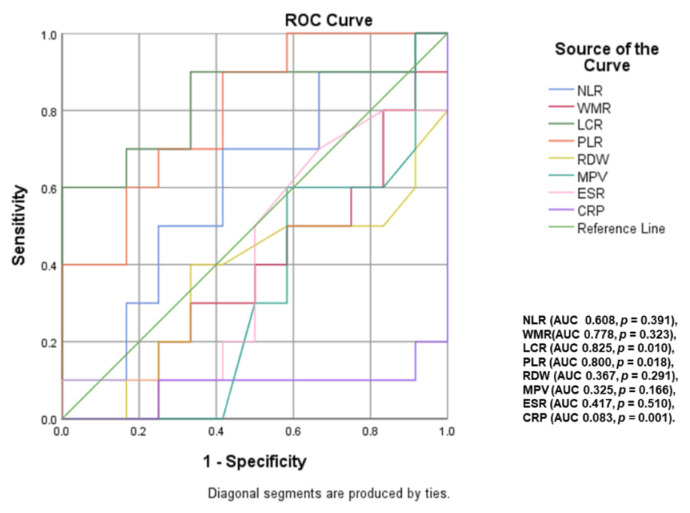
Inflammatory parameters with a predictive role for CPAP adherence in patients with moderate OSA (NLR: neutrophil–lymphocyte ratio; WMR: white blood cell count-to-mean platelet volume ratio; LCR: lymphocyte-to-CRP ratio; ESR: erythrocyte sedimentation rate; CRP: C-reactive protein; PLR: platelet-to-lymphocyte ratio; RDW: red cell distribution width; MPV: mean platelet volume).

**Figure 6 ijms-23-12431-f006:**
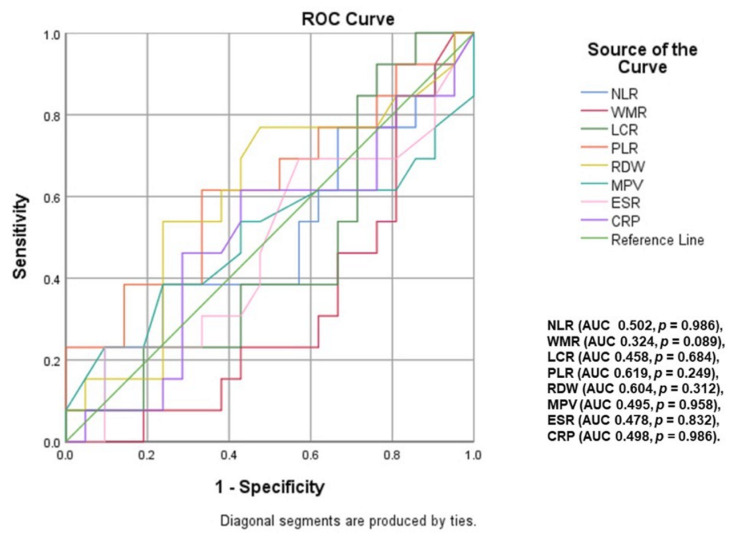
Inflammatory parameters with a predictive role for CPAP adherence in patients with severe OSA (NLR: neutrophil–lymphocyte ratio; WMR: white blood cell count-to-mean platelet volume ratio; LCR: lymphocyte-to-CRP ratio; ESR: erythrocyte sedimentation rate; CRP: C-reactive protein; PLR: platelet-to-lymphocyte ratio; RDW: red cell distribution width; MPV: mean platelet volume).

**Table 1 ijms-23-12431-t001:** Demographic, anthropometric and biological parameters in controls and patients with mild, moderate and severe OSA.

Parameters	Control Group (*n* = 31)	Mild OSA (*n* = 33)	*p** Value	Moderate OSA (*n* = 22)	*p^%^* Value	Severe OSA (*n* = 37)	*p^^^* Value
BMI	32.11 ± 5.16	32.34 ± 5.44	0.894	32.65 ± 6.16	0.762	35.41 ± 5.63	0.035
Age, y	49.55 ± 14.01	54.06 ± 15.37	0.225	57.68 ± 9.18	0.021	58.49 ± 9.49	0.003
Males	16 (51.6%)	20 (60.6%)	0.476	16 (72.6%)	0.126	26 (70.3%)	0.118
Active smokers	5 (16.1%)	4 (12.1%)	0.645	1 (4.5%)	0.190	6 (16.2%)	0.992
Diabetes mellitus	6 (19.4%)	9 (27.3%)	0.453	9 (40.9%)	0.085	10 (27.0%)	0.459
HFrEF	2 (6.45%)	1 (3.03%)	0.515	0 (0%)	0.226	2 (5.4%)	0.857
Statin therapy	19 (61.3%)	31 (93.9%)	0.001	22 (100%)	0.009	28 (75.57%)	0.200
WBC	6637.10 ± 1473.39	6920.30 ± 1731.96	0.485	6322.73 ± 1719.62	0.478	6883.24 ± 1858.72	0.553
RDW	12.94 ± 1.24	12.86 ± 2.73	0.887	13.53 ± 1.03	0.071	13.94 ± 1.35	0.002
MPV	10.35 ± 1.55	10.37 ± 1.06	0.957	9.03 ± 0.80	0.001	9.62 ± 0.96	0.019
Neutrophil	3755.81 ± 1160.16	3900.91 ± 1099.94	0.609	3729.55 ± 1274.41	0.938	4191.35 ± 1527.75	0.197
Lymphocyte	2010.97 ± 472.13	2146.97 ± 704.42	0.371	1877.27 ± 640.81	0.386	1955.14 ± 650.49	0.692
CRP	0.44 ± 0.47	0.51 ± 0.42	0.544	0.61 ± 0.43	0.003	0.98 ± 1.30	0.010
ESR	13.86 ± 11.28	15.25 ± 10.00	0.650	12.55 ± 8.51	0.668	20.17 ± 18.48	0.163
NLR	1.93 ± 0.68	1.90 ± 0.54	0.837	2.10 ± 0.75	0.400	2.34 ± 1.13	0.087
WMR	650.40 ± 161.36	673.22 ± 176.66	0.592	703.43 ± 201.47	0.293	713.61 ± 162.66	0.114
LCR	4370.48 ± 8375.09	1632.83 ± 1421.37	0.078	11,318.78 ± 32,732.56	0.269	17,090.79 ± 53,940.05	0.206
PLR	123.88 ± 40.46	123.97 ± 36.05	0.993	127.79 ± 41.69	0.733	134.45 ± 58.31	0.398

All values are expressed as mean ± standard deviation (SD) or n (%); OSA: obstructive sleep apnea; BMI: body mass index; y: years; HFrEF: heart failure with reduced ejection fraction (<40%); WBC: white blood cells (normal range: 5000–1000/mm^3^ in males, 4000–9000/mm^3^ in females); RDW: red cell distribution width (normal range: 11–16%); MPV: mean platelet volume (normal range 6–10 fl); Neutrophil normal range: 3250–7500/mm^3^ in males, 1800–5000 in females; Lymphocyte normal range: 1250–3500/mm^3^ in males, 1000–3150/mm^3^ in females; CRP: C-reactive protein (normal range 0–1 mg/dL); ESR: erythrocyte sedimentation rate (normal range 4–23 mm/h in males, 4–25 mm/h in females); NLR: neutrophil-to-lymphocyte ratio, normal range 0.43~2.75 in males and 0.37–2.87 in females; WMR: mean platelet volume ratio, normal range—not defined; LCR: lymphocyte-to-C-reactive protein ratio, normal range—not defined; PLR: platelet-to-lymphocyte ratio, normal range 36.63–149.13 in males and 43.36–172.68 in females; *p**—control group vs. mild OSA (calculated using paired samples *t* test); *p^%^*—control group vs. moderate OSA (calculated using paired samples *t* test); *p^^^*—control group vs. severe OSA (calculated using paired samples *t* test).

**Table 2 ijms-23-12431-t002:** Correlations between AHI and inflammatory parameters according to OSA severity.

	All Patients—Controls and OSA (*n* = 123)		Control Group (*n* = 31)		Mild OSA (*n* = 22)		Moderate OSA (*n* = 22)		Severe OSA (*n* = 37)	
	AHI									
	r	*p* Value	r	*p* Value	r	*p* Value	r	*p* Value	r	*p* Value
NLR	0.199	0.027	0.009	0.963	−0.290	0.101	0.383	0.079	−0.006	0.970
WMR	0.260	0.003	0.022	0.908	0.124	0.493	0.247	0.268	0.603	<0.001
LCR	0.166	0.073	−0.474	0.008	−0.037	0.842	0.042	0.854	0.011	0.948
ESR	0.220	0.022	−0.051	0.826	−0.133	0.501	−0.049	0.829	0.154	0.362
CRP	−0.199	0.030	0.221	0.240	0.242	0.191	0.077	0.733	0.389	0.019
PLR	−0.025	0.780	0.701	<0.001	0.607	<0.001	−0.048	0.831	−0.386	0.018
RDW	0.201	0.011	0.191	0.324	−0.183	0.308	0.136	0.547	0.041	0.809
MPV	−0.158	0.019	0.092	0.624	0.131	0.468	−0.107	0.637	0.050	0.767

AHI: apnea–hypopnea index; OSA: obstructive sleep apnea; r: Pearson correlation; NLR: neutrophil–lymphocyte ratio; WMR: white blood cell count to mean platelet volume ratio; LCR: lymphocyte-to-CRP ratio; ESR: erythrocyte sedimentation rate; CRP: C-reactive protein; PLR: platelet-to-lymphocyte ratio; RDW: red cell distribution width; MPV: mean platelet volume. r = Pearson correlation value; *p* value obtained using bivariate correlations (2-tailed).

**Table 3 ijms-23-12431-t003:** Anthropometric and inflammatory parameters, at baseline and after 8 weeks of CPAP (moderate and severe OSA).

Parameters	Baseline (*n* = 59)	Follow-Up (*n* = 59)	*p* Value
Anthropometric parameters			
Weight, kg	101.22 ± 17.32	99.13 ± 17.05	<0.001
BMI, kg/m^2^	34.37 ± 5.89	33.84 ± 5.77	0.001
Inflammatory parameters			
RDW	13.79 ± 1.25	13.98 ± 1.58	0.171
MPV	9.40 ± 0.94	9.30 ± 0.90	0.166
WBC	6674.24 ± 1813.78	6654.24 ± 1795.14	0.932
ESR	17.41 ± 15.72	17.52 ± 13.54	0.804
CRP	0.84 ± 1.06	0.82 ± 0.77	0.902
NLR	2.24 ± 1	2.31 ± 0.86	0.561
PLR	131.96 ± 52.44	136.73 ± 49.54	0.325
WMR	709.81 ± 176.47	718.32 ± 193.77	0.641
LCR	14,863 ± 46,651.03	8635.75 ± 22,383.65	0.373

CPAP: continuous positive airway pressure; n: number; BMI: body mass index; RDW: red cell distribution width; MPV: mean platelet volume; WBC: white blood cell count; ESR: erythrocyte sedimentation rate; CRP: C-reactive protein; NLR: neutrophil–lymphocyte ratio; PLR: platelet-to-lymphocyte ratio; WMR: white blood cell count-to-mean platelet volume ratio; LCR: lymphocyte-to-CRP ratio; *p* value calculated using paired samples *t* test.

**Table 4 ijms-23-12431-t004:** Anthropometric and inflammatory parameters, at baseline and after 8 weeks of CPAP, in non-adherent and adherent subgroups.

	CPAP Non-Adherent (*n* = 25)			CPAP-Adherent (*n* = 34)		
Parameters	Baseline	Follow-Up	*p*	Baseline	Follow-Up	*p*
Anthropometric parameters						
Weight, kg	108.88 ± 17.30	106.14 ± 17.68	<0.001	95.59 ± 15.25	93.82 ± 14.68	0.001
BMI, kg/m^2^	36.46 ± 7.05	35.81 ± 6.94	0.007	32.84 ± 4.36	32.25 ± 4.24	<0.001
Inflammatory parameters						
RDW	13.66 ± 1.25	13.83 ± 1.63	0.684	13.89 ± 1.26	14.09 ± 1.56	0.293
MPV	9.75 ± 0.96	9.43 ± 0.92	0.229	9.14 ± 0.85	9.21 ± 0.88	0.342
ESR	15.60 ± 12.65	15.36 ± 10.90	0.831	18.74 ± 17.71	18.15 ± 15.20	0.339
CRP	0.91 ± 0.83	0.65 ± 0.51	0.05	0.79 ± 1.20	0.95 ± 0.91	0.05
NLR	2.35 ± 1.24	2.27 ± 0.95	0.856	2.18 ± 0.81	2.35 ± 0.81	0.176
PLR	113.26 ± 44.33	116.34 ± 30.02	0.770	145.72 ± 54.27	151.73 ± 55.79	0.93
WMR	763.45 ± 20.6.23	757.62 ± 195.90	0.090	670.38 ± 141.50	689.43 ± 189.89	0.42
LCR	15,103.61 ± 54,917.14	12,161.84 ± 32,819.72	0.83	14,289.96 ± 39,952.93	6071.33 ± 9300.65	0.171

CPAP: continuous positive airway pressure; n: number; BMI: body mass index; RDW: red cell distribution width; MPV: mean platelet volume; WBC: white blood cell count; ESR: erythrocyte sedimentation rate; CRP: C-reactive protein; NLR: neutrophil–lymphocyte ratio; PLR: platelet-to-lymphocyte ratio; WMR: white blood cell count-to-mean platelet volume ratio; LCR: lymphocyte-to-CRP ratio; *p* value calculated using paired samples *t* test (for each group: CPAP non-adherent and CPAP-adherent).

**Table 5 ijms-23-12431-t005:** Multinominal logistic regression analysis results—patients with moderate and severe OSA (*n* = 59).

CPAP Adherence. Variables	B	Std. Error	Wald	df	Sig.	Exp (B)	95% Confidence Interval for Exp (B)	
							Lower Bound	Upper Bound
AHI	−0.004	0.017	0.056	1	0.812	0.996	0.964	1.029
NLR	−0.981	0.450	4.754	1	0.029	0.375	0.155	0.906
PLR	0.021	0.010	4.127	1	0.042	1.021	1.001	1.042
WMR	0.001	0.002	0.077	1	0.782	1.001	0.996	1.005
LCR	0.000	0.000	0.012	1	0.913	1.000	1.000	1.000
Age	0.097	0.044	4.793	1	0.029	1.102	1.010	1.203
BMI	−0.013	0.067	0.037	1	0.846	0.987	0.865	1.126

CPAP: Continuous positive airway pressure; AHI: apnea–hypopnea index; NLR: neutrophil–lymphocyte ratio; WMR: white blood cell count-to-mean platelet volume ratio; LCR: lymphocyte-to-CRP ratio; PLR: platelet-to-lymphocyte ratio; Statistical analysis: multinominal logistic regression analysis using as parameters demographics (age), anthropometric data (BMI), AHI and inflammatory parameters (NLR, PLR, WMR, LCR).

## Data Availability

The data that support the findings of this study are available on request from the corresponding authors.

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
