# Peer review of "CPAP Influence on Readily Available Inflammatory Markers in OSA—A Pilot Study"

_ijms, 2022, doi:10.3390/ijms232012431_

Round 1

Reviewer 1 Report

Comments to the authors:

1. Sleep apnea and its severity HAS to be defined by AHI which are recorded on a sleep study. Using additional inflammatory markers to predict the  same severity which was defined using a sleep study is an additional cost to patient and health care. Even if they are cheap, the test in itself will add unnecessary cost to predict severity which is already obtained using Sleep study. Authors could instead hypothesize to see if these markers predict adherence?

2. What were the chronic severe conditions that were excluded?

3. How many of these patients had Diabetes and if so, did they have a A1C level. Because high A1C in itself causes proinflammatory state.

4. This study would benefit from presenting data on smoking status such as never, former or active along with defining pack per years smoking history. 

5. How many patients had heart failure?

6. Is the data/result different if the outcomes were measured after matching for age given that age was statistically significant in demographics in moderate and severe OSA, compared to mild and control.

7. Were any of these patients on steroids or statin therapy?

Thank you.

Reviewer 2 Report

The authors retrospectively show that some inflammatory markers are correlated with OSA and elaborate on possible usage of these in screening or for risk stratification in patients. A thorough literature search was performed and appropriate inclusion and exclusion criteria were defined and applied.

The number of 123 Patients included seems sufficient for a pilot study though it leaves only small numbers of patients in the subgroups.

I have a few remarks:

 In line 104 the CRP value for mild OSA is 2.57, in Table 1 below the number is 0.51.

 Also it would facilitate the evaluation of the study if the standard values of the blood parameters would be included in Table 1.

 In line 169 PLR is shown as statistically relevant with p = 0.510.

Round 2

Reviewer 1 Report

Thank you for the edits.

Author Response

Thank you